# ADAM17 Is an Essential Factor for the Infection of Bovine Cells with Pestiviruses

**DOI:** 10.3390/v14020381

**Published:** 2022-02-13

**Authors:** Marianne Zaruba, Hann-Wei Chen, Ole Frithjof Pietsch, Kati Szakmary-Braendle, Angelika Auer, Marlene Mötz, Kerstin Seitz, Stefan Düsterhöft, Aspen M. Workman, Till Rümenapf, Christiane Riedel

**Affiliations:** 1Institute of Virology, Department of Pathobiology, University of Veterinary Medicine, 1210 Vienna, Austria; Marianne.Zaruba@vetmeduni.ac.at (M.Z.); Hann-Wei.Chen@vetmeduni.ac.at (H.-W.C.); 1622231@students.vetmeduni.ac.at (O.F.P.); Kati.Szakmary-Braendle@vetmeduni.ac.at (K.S.-B.); angelika.auer@vetmeduni.ac.at (A.A.); marlene.moetz@vetmeduni.ac.at (M.M.); kerstin.seitz@vetmeduni.ac.at (K.S.); Till.ruemenapf@vetmeduni.ac.at (T.R.); 2Institute for Molecular Pharmacology, RWTH Aachen University, 52062 Aachen, Germany; sduesterhoeft@ukaachen.de; 3US Meat Animal Research Center, United States Department of Agriculture, Agricultural Research Service, Clay Center, NE 68901, USA; aspen.workman@usda.gov

**Keywords:** pestivirus, bovine viral diarrhea virus, cells resistant to BVDV infection, ADAM17

## Abstract

The entry of BVDV into bovine cells was studied using CRIB cells (cells resistant to infection with bovine viral diarrhea virus [BVDV]) that have evolved from MDBK cells by a spontaneous loss of susceptibility to BVDV. Recently, larger genetic deletions were reported but no correlation of the affected genes and the resistance to BVDV infection could be established. The metalloprotease ADAM17 was reported as an essential attachment factor for the related classical swine fever virus (CSFV). To assess whether ADAM17 might be involved in the resistance of CRIB-1 cells to pestiviruses, we analyzed its expression in CRIB-1 and MDBK cells. While ADAM17 protein was detectable in MBDK cells, it was absent from CRIB-1 cells. No functional full-length ADAM17 mRNA could be detected in CRIB cells and genetic analysis revealed the presence of two defective alleles. Transcomplementation of functional ADAM17 derived from MDBK cells in CRIB-1 cells resulted in a nearly complete reversion of their resistance to pestiviral infection. Our results demonstrate that ADAM17 is a key cellular factor for the pestivirus resistance of CRIB-1 cells and establishes its essential role for a broader range of pestiviruses.

## 1. Introduction

Pestiviruses (family *Flaviviridae*) are small, enveloped viruses with a single stranded, positive sense RNA genome of about 12.3 kb. Pestiviral particles harbor three glycoproteins in their envelope, E^rns^, E1 and E2. The glycoproteins E1 and E2 are usually present as a heterodimer in the virus particle [1,2], and E2 is responsible for receptor binding and together with E^rns^ a target of neutralizing antibodies [1,3,4]. Bovine viral diarrhea virus (BVDV, *Pestivirus A*) and classical swine fever virus (CSFV, *Pestivirus C*) are two important members of the genus and are O.I.E. (World Organisation for Animal Health) listed and notifiable in many countries. They both cause severe economic problems due to reduced animal performance, cost of disease prevention and treatment [5,6] or re-emergence [7,8].

Pestivirus entry likely commences by an initial attachment to cellular glycosaminoglycans [9,10,11]. Several cell surface proteins have been identified as mediators of pestivirus entry. For BVDV and atypical porcine pestivirus (APPV), CD46 was discovered as a receptor molecule that interacts with E2 [12,13,14]. For CSFV, the ADAM17 [15] surface metalloprotease was recently reported to act as an entry factor via interaction with E2. Other receptor candidates for CSFV are annexin 2 [16], MERTK [17], and laminin receptor (LMNR) [18]. After binding to a receptor molecule, pestivirus entry proceeds, as experimentally shown for BVDV and CSFV, by clathrin-mediated endocytosis [19,20,21,22] that depends on the presence of cholesterol and functional dynamin. After clathrin-mediated uptake, CSFV particles colocalize with RAB5-(Ras-like GTPases) and RAB7-positive endosomes [21], indicating that pestiviral fusion takes place in the low pH compartment of late endosomes or lysosomes. Within the endosomal compartment, studies employing BVDV demonstrated that low pH is a prerequisite to trigger fusion activity. However, BVDV particles are acid-resistant and fusion with the cell membrane can only be induced in vitro in the presence of low pH and reducing substances, such as DTT [19]. This strongly implies the need for reduction of disulfide bonds and potentially even proteolytic processing—as, for example, required for Ebola virus entry [23,24]—to acquire fusion competence.

There is evidence that factors exist which inhibit the entry of several pestiviruses. Studies characterizing the CRIB-1 subclone of the MDBK cell line [25,26] demonstrated its resistance to BVDV and CSFV. CRIB-1 cells spontaneously occurred as survivors from an MDBK cell culture during an infection with the cytopathic BVDV NADL strain. Upon transfection of viral genomes, CRIB-1 cells are permissive for pestivirus replication and can produce infectious progeny, hence their resistance to pestivirus infection is likely evoked by a modification inhibiting virus entry. No difference in susceptibility for other bovine and porcine viruses between CRIB-1 cells and the parental MDBK cell line could be detected. A recent study comparing the genomes of MDBK and CRIB-1 cells discovered compound heterozygous deletions in three genes, protein tyrosine phosphatase non receptor type 12 protein (*PTPN12*), glutamate ionotropic receptor delta type subunit 2 protein (*GRID2*) and RAB GTPase activating protein 1 like protein (*RABGAP1L*) [27]. Neither single nor triple knock-out of these genes affected the susceptibility of the parental MDBK cell line to BVDV, indicating that these profound genomic alterations are not (directly) responsible for the resistance of CRIB-1 cells to BVDV infection.

ADAM17 belongs to the a disintegrin and metalloproteinase (ADAM) family of proteins. These proteins are involved in ectodomain shedding of cell surface proteins, in the case of ADAM17 for example tumor necrosis factor [28]. They are type-I transmembrane proteins and their catalytic consensus motive is HEXGEHXXGXXH [29]. The activity of ADAM17 is regulated at multiple levels. The properly folded, mature protein is on its own retained in the ER of the cell and needs the interaction with rhomboid proteins to reach the cell surface [30,31]. Once on the cell surface, the protease inhibitor TIMP3 can induce ADAM17 inactivation [32], potentially associated with dimer formation [33], whilst an increased activity is induced by phosphorylation of the C-terminal cytoplasmatic domain by MAP kinases [34,35,36]. For CSFV, a direct interaction of E2 with the ADAM17 protease domain could be shown which depends on the Zn^2+^ cation [15], as demonstrated by a change in the Zn^2+^ binding motif (H405D) and depletion with the Zn^2+^ chelating agent 1′10 phenanthroline. Also, an inhibitory effect of TIMP3 on the susceptibility to CSFV was recorded, indicating a potential competition for ADAM17 binding. In contrast, deletion of the ADAM17 C-terminal, PACS-2 interacting region did not reduce the ability of ADAM17 to mediate CSFV entry, indicating that it is responsible for binding of CSFV particles via E2, but not (directly) involved in virus uptake by clathrin mediated endocytosis.

The aim of this study was to elucidate the role of ADAM17, a recently discovered essential entry factor of CSFV [15], in the pestivirus resistance of CRIB-1 cells.

## 2. Materials and Methods

### 2.1. Cell Culture

MDBK [37] (ATCC CCL-22), HEK293 [38] (ATCC CRL-1573), SK6 [39] (kindly provided by the collection of cell lines in veterinary medicine of the Friedrich Löffler Institute, Germany) and CRIB-1 [25] (a kind gift of Ruben Donis) cells were grown at 37 °C, 5% CO_2_ and 100% humidity. Full medium for all these cell lines consisted of DMEM high glucose (Biowest, Nuaillé, France) supplemented with 100 U/mL penicillin, 100 µg/mL streptomycin and 10% fetal calf serum (Corning, Tewksbury, MA, USA). The fetal calf serum was tested extensively by cell culture passaging, immunofluorescence, and RT-PCR for freedom from pestiviruses. Viruses employed in this study included CSFV Alfort Tübingen [40] (subtype 2.3 [41]), Linda virus (LindaV) [42], a HoBi-like pestivirus (HoBiPeV) strain (Gi2011_1) isolated from a serum contamination [43] and the BVDV strains NADL [4] and C87 (both subtype 1A [4]). CSFV and LindaV were propagated in SK6 cells, whilst the other viruses were propagated in MDBK cells.

### 2.2. RNAseq Analysis of MDBK and CRIB-1 Cells and Validation

Cellular RNA was extracted in triplicates from MBDK and CRIB cells with the Qiagen RNeasy Mini kit. RNA integrity was assessed on a 4200 TapeStation with the RNA ScreenTape assay (Agilent, Santa Clara, CA, USA). In total, 750 ng total RNA were used for mRNA library preparation with the SENSE mRNA Library Prep Kit V2 (Lexogen, Vienna, Austria) according to the manufacturer’s protocol. Library quality control was completed with the D1000 ScreenTape kit on the 4200 TapeStation (Agilent, Santa Clara, CA, USA). All samples were sequenced on one lane of an Illumina HiSeq platform, implementing single-end 50-bp reads, using HiSeq SBS v4 reagents (Illumina, San Diego, CA, USA). Sequencing was completed at the next generation sequencing unit of the Vienna Biocenter Core Facilities (VBCF, Vienna, Austria). Resulting data were analyzed with the Qiagen Genomics Workbench (Qiagen, Hilden, Germany).

The amounts of selected mRNAs either reported to be present in similar amounts of strongly differing amounts in the RNAseq analysis between CRIB and MDBK cells was confirmed by SYBR green based RT-qPCRs (Luna Universal One-Step RT-qPCR Kit, NEB, Ipswich, MA, USA). A probe-based beta-actin qPCR [44] was employed to normalize for the RNA input (Luna Universal Probe One-Step RT-qPCR Kit, NEB, Ipswich, MA, USA), taking the different PCR efficiencies as determined by serial dilutions into consideration. Primers employed in the qPCR assays can be found in Appendix A. qPCRs were performed on a Jena qTOWER^3^ (Jena Bioscience, Jena, Germany).

### 2.3. Cloning of ADAM17 cDNA from MDBK and CRIB-1 Cells and Expression of ADAM17 from MDBK Cells in CRIB-1 Cells

Cellular RNA was extracted from MDBK or CRIB-1 cells with the NEB Monarch RNA extraction kit and reverse transcribed with primer 3′ ADAM17 RT rev using the M-MulV RT (NEB, Ipswich, MA, USA). cDNA was purified using the Monarch PCR purification kit (NEB, Ipswich, MA, USA) and employed in an initial PCR amplification step with primers 5′ ADAM17 for and 3′ ADAM17 rev. One µL of this PCR reaction was employed in a second PCR reaction employing primers 5′ ADAM17 for cloning and 3′ ADAM17 rev cloning, and the resulting PCR product was analysed on a gel and the corresponding band excised and purified with the Monarch gel purification kit (NEB, Ipswich, MA, USA). A retroviral expression vector encoding a flag tag, a P2A cleavage site and a puromycin resistance gene directly C-terminal of the ADAM17 integration site [45] was amplified by PCR employing the primers 5′ ADAM17 rev cloning and 3′ ADAM17 for cloning, the product was purified as described above and the fragments were ligated with the HifiDNA Assembly mix (NEB, Ipswich, MA, USA) and transfected in *E. coli* HB101. The resulting constructs were validated by restriction enzyme digest and sequence analysis (Eurofins Genomics, Ebersberg, Germany). ApE (http://biologylabs.utah.edu/jorgensen/wayned/ape/, accessed on 9 February 2022) was used for sequence comparison, primer and plasmid design. The SH3BGRL cDNA was cloned following the same approach employing the primers as shown in Appendix A.

Retroviral pseudotypes were generated as described in [46]. In total, 2.5 × 10^4^ CRIB-1 cells or MBDK cells were transduced with the lentiviral pseudotypes in one well of a 24 well plate (Starlab, Hamburg, Germany) each. Then, 72 h after transduction, the cells were transferred to a 10 cm dish (Starlab, Hamburg, Germany) and selected based on their resistance to puromycin by adding 2 µg/mL puromycin (Merck, Kenilworth, NJ, USA) to the medium. The surviving cells were propagated as a polyclonal population.

### 2.4. Genetic Characterization of ADAM17

Publicly available reads of a full genome sequencing experiment of CRIB-1 and MDBK cells (Bioproject PRJNA761701) performed in Workman et al. [27] were analyzed for changes in the ADAM17 gene employing the program IGV [47,48].

### 2.5. Detection of ADAM17 Expression

ADAM17 expression was detected by Western blot analysis employing a commercially available anti ADAM17 serum previously described to cross react with porcine ADAM17 (ab39162, Abcam, Cambridge, UK). Western blot analysis was performed as described in [49] and the primary antibody was employed at a dilution of 1:1000 in PBS 0.1% Tween20 (*v*/*v*) containing 4% skimmed milk powder. For enrichment of glycosylated proteins, ADAM17 was precipitated with Concanavalin A sepharose beads (Merck, Kenilworth, NJ, USA). Cells were detached from a confluent well of a 6-well plate (Starlab, Hamburg, Germany) by tryptic digestion, pelleted (2 min 400× *g*), resuspended in 1 mL lysis buffer (50 mM Tris, 137 mM NaCl, 2 mM EDTA, 1% Triton X-100) supplemented with protease inhibitor cocktail (Carl Roth, Karlsruhe, Germany) and 10 mM 1,10-Phenanthrolin (Merck, Kenilworth, NJ, USA) and incubated for 1 h at 4 °C on a spinning wheel. After spinning for 20 min at 13,000 rpm in a tabletop centrifuge at 4 °C, the supernatant was removed and incubated with 20 µL of bead slurry (Merck, Kenilworth, NJ, USA) on a spinning wheel at 4 °C overnight. Beads were washed five times with 1 mL lysis buffer 45 s 350× *g*, resuspended in 60 µL loading buffer and incubated for 5 min at 95 °C before the sample was being subjected to Western blot analysis.

Expression of ADAM17 in HEK293 cells heterologously expressing ADAM17 derived from MDBK cells was confirmed by detection with an anti-flag monoclonal antibody (M2, Merck, Kenilworth, NJ, USA).

### 2.6. Determination of Susceptibility

In total, 1 × 10^5^ MDBK, MDBK_ADAM17_, MDBK_SH3BGRL_, CRIB, CRIB_ADAM17_ or CRIB_SH3BGRL_ cells/well of a 24-well plate were seeded and subsequently infected with serial 10-fold dilutions of BVDV-1 strain NADL [4], BVDV-1 strain C87, BVDV-1 strain C87 labelled with mClover at the E2 N-terminus, HoBiPeV [43], CSFV strain Alfort Tübingen [40] and Linda virus [42]. Four hours after infection, the medium was replaced with full medium containing 1% carboxymethylcellulose (Carl Roth, Karlsruhe, Germany) without phenolred. After 24 h (NADL), 48 h (C87, BVDV3, LindaV) or 72 h (CSFV), the cells were washed and fixed with methanol acetone 1:1. Antibody A18 [3] was employed for the detection of CSFV, whilst all other viruses were detected with antibody 6A5 [50]. Antibodies were employed as unpurified hybridoma cell supernatant at a dilution of 1:10 in PBS 0.1% Tween20. A Cy3-labelled goat anti-mouse antibody (Dianova, Hamburg, Germany) was used as a secondary antibody. Focus forming units (FFU) were quantified on an Olympus IX70 fluorescence microscope (OLYMPUS, Hamburg, Germany) and the susceptibility in percent was calculated as follows:Susceptibility = (FFU/mL tested cell line)/(FFU/mL MDBK cells) × 100.

Cellular nuclei were stained by incubation with DAPI 1 µg/mL in PBS for 5 min at room temperature.

### 2.7. Determination of Spread

In total, 2 × 10^5^ MDBK, MDBK_ADAM17_, MDBK_SH3BGRL_, CRIB, CRIB_ADAM17_ or CRIB_SH3BGRL_ were seeded in each well of a 12-well plate. The cells were infected with an MOI of 0.001 with a C87 clone encoding mClover at the N-terminus of E2 and incubated for 72 h. Subsequently, cells were detached with trypsin and fixed with 4% PFA in PBS for 20 min at 4 °C. Cells were washed and resuspended in PBS 1% FCS and the percentage of mClover positive cells was determined in an Amnis Flow Sight instrument (Luminex, Austin, TX, USA).

## 3. Results

### 3.1. ADAM17 Alleles Are Disrupted in CRIB-1 Cells

An essential role of ADAM17 for CSFV entry has recently been demonstrated by Yuan et al. [15]. Whilst analyzing the transcriptomic differences of MDBK and CRIB-1 cells, we observed a 3-fold reduced number of transcripts of ADAM17 in CRIB-1 cells compared to MDBK cells in an RNAseq experiment. SYBR Green based RT-qPCR analysis confirmed this result, reporting a 6-fold decreased amount of ADAM17 mRNA in CRIB cells. Additionally, the mRNA of an inhibitor of ADAM17, TIMP3, was upregulated 3-fold in CRIB-1 cells in the RNAseq analysis and 5-fold increased in RT-qPCR.

The increased mRNA level of TIMP3 and the decreased mRNA level of ADAM17 in CRIB-1 cells were reminiscent of the findings by Yuan et al. [15], who reported a correlation of cellular susceptibility to CSFV and the levels of ADAM17 and TIMP3 expression. To validate whether the observed change in mRNA expression levels resulted in a different abundance of ADAM17 in the cells, we performed a Western blot analysis employing a commercially available polyclonal antibody generated against a peptide in the C-terminus of human ADAM17. This antibody has previously been reported to interact with porcine ADAM17 [15]. In lysates of MDBK cells, three bands of molecular weights of approximately 85 kD, 95 kD and 120 kD could be detected. In CRIB-1 cell lysates, only the 85 kD band was detectable (Figure 1A). The 95 kD and the 120 kD band correspond to the glycosylated mature form and the glycosylated pro form of ADAM17, indicating the absence of ADAM17 in CRIB-1 cells. The origin of the 85 kD band is unknown and attributed to unspecific binding.

To confirm the identity of the 95 kD and 120 kD bands, glycosylated proteins were precipitated from the lysates of CRIB-1 and MDBK cells with Concanavalin A beads (ConA) and bound protein was again analyzed in Western blot. No band could be detected in the sample derived from CRIB-1 cells, whilst the 95 kD band and the 120 kD band were present in the sample derived from MDBK cells (Figure 1A). Therefore, no glycosylated ADAM17 forms can be detected in CRIB-1 cells.

To genetically define the CRIB-1 phenotype observed upon analysis of ADAM17 by Western blot, we sequenced the full-length ADAM17 cDNA of MDBK and CRIB-1 cells. The coding region is 2478 nucleotides (825 amino acids). In total, 15 out of 15 sequenced clones containing the CRIB-1 derived ADAM17 cDNA encoded a nucleotide deletion at position 932, resulting in a premature stop codon at ADAM17 amino acid 311. This deletion was not detected in cDNA derived from MDBK cells. Analysis of publicly available full genome sequencing reads of CRIB-1 and MDBK cells focusing on the *ADAM17* gene confirmed the presence of the one nucleotide deletion in one allele and revealed a large deletion of 180 kb (nt 87955976-88136128 on chromosome 11) starting between exon 14 and 15 of the *ADAM17* gene (Figure 2), also on one allele only. In combination with the analysis of the ADAM17 cDNA, these results demonstrate the presence of two defective *ADAM17* alleles in CRIB-1 cells. The absence of ADAM17 in CRIB-1 cells is therefore caused by genetic defects in the *ADAM17* gene.

### 3.2. ADAM17 Transcomplementation Reverts the Phenotype of CRIB-1 Cells for a Diverse Array of Pestiviruses

To functionally assess whether ADAM17 was indeed a driving force behind the pestivirus resistance of CRIB-1 cells, we cloned the ADAM17 cDNA from MDBK cells in a retroviral vector also expressing a flag tag and a puromycin resistance in the same open reading frame. To control for the effect of the heterologous expression on the pestiviral resistance of CRIB-1 cells and the susceptibility of MDBK cells, we also included SH3BGRL (SH3 Domain Binding Glutamate Rich Protein Like). Its mRNA was not detectable by RT-qPCR in CRIB-1 cells but was detectable in MDBK cells and it was therefore chosen as a control protein differentially expressed in CRIB-1 and MDBK cells. The expression of proteins from the resulting plasmids was confirmed by Western blot analysis of transfected HEK293 cells (Appendix A) and the expression of ADAM17 in CRIB-1 cells was confirmed by ADAM17 specific Western blot analysis (Figure 1B).

MDBK cells and CRIB-1 cells were transduced with lentiviral pseudotypes and selected based on their resistance to puromycin, resulting in MDBK_ADAM17_, MDBK_SH3BGRL_, CRIB-1_ADAM17_ and CRIB-1_SH3BGRL_. Subsequently, the susceptibility of the different cell lines to infection with two BVDV-1 strains (NADL and C87), one CSFV strain (Alfort Tübingen), one HoBiPeV isolate and one Linda virus isolate was tested. A near complete reversion of the pestivirus resistance of CRIB-1 cells could be observed when ADAM17 was expressed from the retroviral vector, whilst no effect could be detected upon expression of SH3BGRL (Figure 3). Specifically, the susceptibility of CRIB-1_ADAM17_ in comparison to MDBK cells was on average 47% (*n* = 4) for BVDV-1 NADL, 40% for BVDV-1 C87, 21% for CSFV Alfort Tübingen, 43% for a HoBiPeV isolate and 26% for LindaV. These data support the necessity of ADAM17 for the entry of a broad array of phylogenetically diverse pestivirus species.

To assess whether additional factors might play a role in the phenotypic resistance of CRIB-1 cells to pestivirus infection, we assessed the effect of ADAM17 expression on the propagation of a fluorophore labelled BVDV clone (Figure 4) [51]. The presence of the fluorophore at the E2 N-terminus did not change the susceptibility of CRIB-1_ADAM17_ compared to the non-labelled, parental virus (Appendix A). MDBK cells, CRIB-1 cells or the respective cells expressing ADAM17 or SH3BGRL were infected with an MOI of 0.001 and analyzed 72 h post infection to determine the amount of fluorophore positive cells in a flow cytometer.

Infection of MDBK_ADAM17_ cells did result in a slightly increased amount of virus positive cells (27.3%) when compared to the parental cell line (23.3%), whilst the amount of infected MDBK_SH3BGRL_ cells was similar to the parental cell line (23.4%). No infected cells could be detected in CRIB-1 cells or CRIB-1_SH3BGRL_. In total, 5.4% of CRIB-1_ADAM17_ cells were infected, resulting in a 5-fold reduced propagation efficiency when compared to MDBK cells.

## 4. Discussion

CRIB-1 cells were an enigma in pestivirus research for several decades. The underlying cause of their resistance to pestivirus infection remained unknown and no genetic change responsible for their pestivirus resistance was identified by full genome sequencing [27]. Here we put forward three lines of evidence that ADAM17 is the missing factor causing the resistance of CRIB-1 cells to pestivirus infection. 1. ADAM17 protein is not detectable in CRIB-1 cells, 2. the ADAM17 gene in CRIB-1 cells is defective and 3. an efficient gain of function by expression of functional ADAM17 from MDBK cells.

To elucidate a role of differentially transcribed mRNAs, we performed an RNAseq experiment, comparing the transcriptomes of CRIB-1 and MDBK cells. Interestingly, we found a decreased amount of ADAM17 mRNA in CRIB-1 cells, which—together with the recent report of the role of ADAM17 for CSFV entry [15]—suggested that ADAM17 might play a role in the pestivirus resistance of CRIB-1 cells. ADAM17 could not be detected in CRIB-1 cells by Western blot analysis. Analysis of the amino acid coding part of the ADAM17 mRNA revealed the presence of a one nucleotide deletion (nt 932), which results in a premature stop codon at amino acid 311. Further characterization of the full *ADAM17* gene with its 19 exons employing the data generated by Workman et al. [27] confirmed the one nucleotide deletion in one allele and revealed a large deletion starting between exon 14 and 15 of the *ADAM17* gene. As no mRNA lacking the deletion of nt 932 could be detected in CRIB-1 cells, the large deletion has to be present in the other allele. Once ADAM17 was provided in trans in CRIB-1 cells, their resistance to infection with a diverse array of pestiviuses (BVDV-1, HoBiPeV, CSFV, LindaV) was nearly completely reverted. Therefore, an essential role of ADAM17 in the entry of pestiviruses and in the resistance of CRIB-1 cells to pestivirus infection can be assumed.

Albeit the stunning effect of ADAM17 transcomplementation on the susceptibility of CRIB-1 cells to pestiviruses, it was never possible to reach the exact same level of susceptibility (still reduced by 50–80%) or propagation efficiency (reduced 5-fold) as observed for MDBK cells. Therefore, additional factors with a less pronounced effect might be lacking or, conversely, might be more abundantly expressed in CRIB-1 cells relative to MDBK cells and affect their phenotype. One interesting candidate is TIMP3, whose expression was increased in CRIB-1 cells in comparison to MDBK cells, as TIMP3 is a specific inhibitor of ADAM17 and its binding site should compete with the interaction of E2 and ADAM17 as mapped by Yuan et al. [15]. In this context it would also be interesting to analyze whether the described deletions of *PTPN12*, *GRID2* and *RABGAP1L* [27] or the heterozygous defect of the *ASAP2* gene, caused by the large deletion starting in the *ADAM17* gene, are responsible for this effect. Comparison of the transcriptomes of MDBK and CRIB cells has revealed a number of possible candidates that need to be tested one by one. Nevertheless, CRIB-1 cells and the particular MDBK cell clone used in our laboratory diverge by almost 30 years, increasing the likelihood that changes acquired at the genetic and transcriptomic level have been acquired after the selection event. Additionally, the utilization of a polyclonally selected population of cells in our experimental setup might result in differences in the ADAM17 expression levels. This heterogeneity has the potential to affect the susceptibility and the spread of BVDV in our experimental setup. Further experiments with ADAM17 knock-out cells and ADAM17 knock-in as well as clonally selected ADAM17 transcomplemented CRIB-1 cells are therefore needed for clarification. Whether ADAM17 is also a prerequisite for cell entry of distantly related pestiviruses, such as APPV, still needs to be evaluated. In our experimental setup, the growth performance of APPV was too poor to allow for an experimental evaluation. Therefore, further experiments are clearly needed in porcine ADAM17 knock-out cells to establish the role of ADAM17 for phylogenetically distant pestiviruses.

When checking for susceptibility, we employed two different BVDV-1 strains, NADL and C87. NADL does not possess a mutation favoring the interaction with cell surface glycosaminoglycans (G479R) in E^rns^, whilst C87 does. This mutation has also been shown to increase CD46-independent entry [52]. ADAM17 expression in CRIB-1 cells had a similar effect on susceptibility to these two virus strains, and it seems therefore likely that ADAM17 has a central role in pestiviral entry independent of which (known) attachment factors are utilized.

Yuan et al. [15] reported the intriguing finding that ADAM17′s protease activity and its proteolytic domain seem essential for CSFV entry [15], which clearly warrants further research to assess the relevance for other pestiviruses and the potential of proteolytic processing of pestiviral surface glycoproteins by ADAM17.

## Figures and Tables

**Figure 1 viruses-14-00381-f001:**
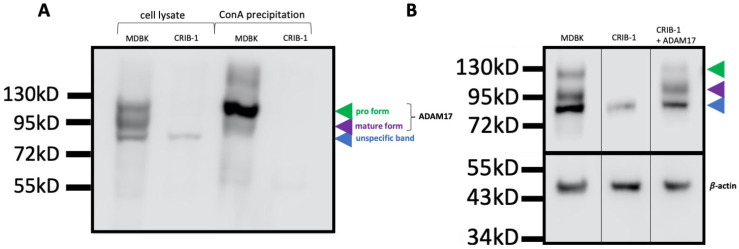
Glycosylated forms of ADAM17 are absent in CRIB-1 cells. (**A**) Western blot analysis of ADAM17 in the cell lysate of MDBK and CRIB-1 cells and in the cellular glycoprotein fraction of MDBK and CRIB-1 cells generated by precipitation with Concanavalin A (ConA) beads. The different bands are indicated by arrow heads. In CRIB-1 cells, only the band designated as unspecific, as its origin is unknown, can be discerned in cell lysate, whilst no band can be detected when glycoproteins are precipitated with ConA beads. (**B**) Detection of ADAM17 and β-actin by Western blot analysis of cell lysates of MDBK, CRIB-1 and CRIB-1 cells expressing ADAM17 from a lentiviral vector (+ADAM17). Upon transcomplementation, the glycosylated pro- and mature form of ADAM17 can be detected in CRIB-1 cells, indicating that the pathways responsible for glycosylation and processing of ADAM17 are intact in CRIB-1 cells.

**Figure 2 viruses-14-00381-f002:**
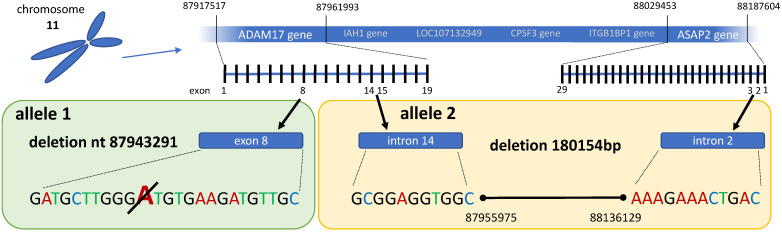
Genetic alterations of the *ADAM17* gene in CRIB-1 cells. Nucleotide positions relative to the ARS-UCD1.2 reference bovine genome. Alignments of reads in the modified regions are shown in Appendix A.

**Figure 3 viruses-14-00381-f003:**
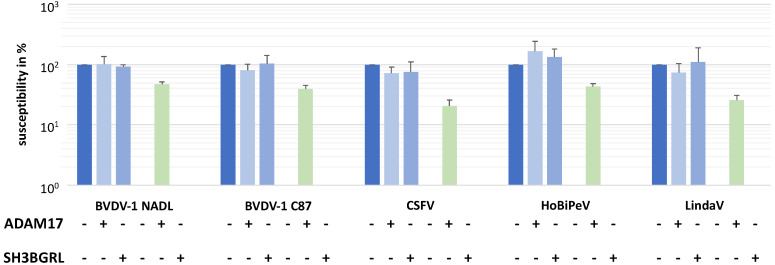
Transcomplementation of bovine ADAM17 renders CRIB-1 cells susceptible to pestiviruses. Susceptibility of MDBK cells (blue) or CRIB-1 (green) cells with (+) or without (−) transcomplementation of ADAM17 or SH3BGRL to different pestiviruses in % of the susceptibility of MDBK cells. Shown are mean and standard deviation of four independent experiments.

**Figure 4 viruses-14-00381-f004:**
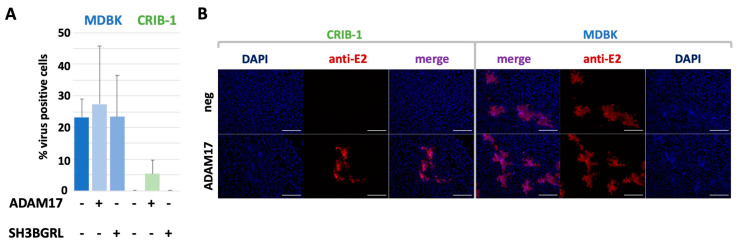
Effect of ADAM17 transcomplementation on BVDV spread. (**A**) Analysis of BVDV spread by flow cytometry. The amount of infected MDBK or CRIB-1 cells with (+) or without (−) transcomplementation of ADAM17 or SH3BGRL was quantified by flow cytometry 72 h after infection with BVDV C87 (E2-mClover) with an MOI of 0.001. Shown are mean and standard deviation of three independent experiments. (**B**) BVDV plaque morphology on MDBK or CRIB-1 cells with (ADAM17) or without (neg) transcomplementation of bovine ADAM17 after staining with the anti-E2 antibody 6A5. Cell nuclei were stained with DAPI. Scale bar = 200 µm.

## Data Availability

Data generated during the RNAseq experiment mentioned in this manuscript are not made publicly available due to ongoing analysis of the data.

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
