# Peer review of "ADAM17 Is an Essential Factor for the Infection of Bovine Cells with Pestiviruses"

_viruses, 2022, doi:10.3390/v14020381_

Round 1

Reviewer 1 Report

Viruses (ISSN 1999-4915)

Manuscript ID viruses-1589460

„ADAM17 is an essential factor for the infection of bovine cells with pestiviruses“ by Marianne Zaruba, Hann-Wei Chen, Ole Frithjof Pietsch, Kati Szakmary-Braendle, Angelika Auer, Marlene Mötz, Kerstin Seitz, Aspen M Workman, Stefan Düsterhöft, Till Rümenapf and Christiane Riedel.

The study of Zaruber et al. demonstrates that the lack of ADAM17 is one reason causing the resistance of CRIB-1 cells to pestiviral infection. The authors first demonstrated by genetic analysis that both ADAM17 alleles were defective in CRIB-1 cells and that no functional full-length ADAM17 mRNA could be detected in these cells. Consequently, ADAM17 protein expression was found to be defective in CRIB-1 but not in MDBK cells. Importantly, retroviral transcomplementation of functional bovine ADAM17 in CRIB-1 cells reversed their resistance to pestiviral infection. Together the results presented in this paper demonstrate that ADAM17 is a key cellular factor for the entry process of pestiviruses into their host cells and establishes this essential role during entry for a broad range of pestiviruses.

The data are clear and convincing and most claims are sufficiently supported by data. This work is technically sound and it provides further proof for the role of ADAM17 during the pestiviral life cycle. Overall, I feel this work work should be published in Viruses. I have outlined a few points for consideration to improve the manscript. They are specified below.

While the effect of the ADAM17 transcomplementation is convincing, one reason for the incomplete susceptibility of CRIB-1ADAM17 cells to pestiviruses could be an inefficient transport of the FLAG-ADAM17 protein to the cell surface after retroviral expression. It would be informative to compare its surface localization in MDBK and CRIB-1ADAM17 cells.

Another reason for the incomplete rescue of the susceptibility of the CRIB-1ADAM17 cells for pestviruses could be the reduced replication capacity in these cells. To address this concern, the authors should compare RNA replication between CRIB-1ADAM17 and MDBK cells after RNA electroporation of in vitro transcribed BVDV full-length or replicon RNAs.

Figure 1A: Could be improved by providing a loading control in the western blot and a control for successful ConA precipitation of a gycosylated control protein from CRIB-1 cells.

Author Response

Please find our answers to the comments of reviewer 1 below. The comments are written in black, our answers in blue, and passages adapted in the text are in blue and italics. 

The data are clear and convincing and most claims are sufficiently supported by data. This work is technically sound and it provides further proof for the role of ADAM17 during the pestiviral life cycle. Overall, I feel this work work should be published in Viruses. I have outlined a few points for consideration to improve the manscript. They are specified below.

We sincerely thank this reviewer for his/her comments and suggestions to improve our manuscript!

While the effect of the ADAM17 transcomplementation is convincing, one reason for the incomplete susceptibility of CRIB-1ADAM17 cells to pestiviruses could be an inefficient transport of the FLAG-ADAM17 protein to the cell surface after retroviral expression. It would be informative to compare its surface localization in MDBK and CRIB-1ADAM17 cells.

We thank the reviewer for this suggestion. For mouse and human ADAM17, it has been demonstrated that different small tags at the C-terminus do not affect ADAM17 surface localisation (doi: 10.1016/j.bbrc.2011.10.056, 10.1074/jbc.M114.557322, 10.1016/j.febslet.2012.03.012). Therefore, we consider it unlikely that the surface localization is responsible for the observed effect. We will however keep this in mind for future experiments!

Another reason for the incomplete rescue of the susceptibility of the CRIB-1ADAM17 cells for pestviruses could be the reduced replication capacity in these cells. To address this concern, the authors should compare RNA replication between CRIB-1ADAM17 and MDBK cells after RNA electroporation of in vitro transcribed BVDV full-length or replicon RNAs.

Experiments evaluating the generation of progeny virus by CRIB-1 cells after electroporation were performed by Flores and Donis 1995 (doi: 10.1006/viro.1995.1187), demonstrating no difference in virus output. These data are in accordance with our own unpublished pilot experiments confirming the comparable virus output after RNA transfection. As shown in Fig. 4A, transcomplementation of ADAM17 does not negatively affect virus spread in MDBK cells. Therefore, we consider a reduced replication capacity in these cells highly unlikely.

Figure 1A: Could be improved by providing a loading control in the western blot and a control for successful ConA precipitation of a gycosylated control protein from CRIB-1 cells.

We thank the reviewer for this suggestion! As a loading control for the cell lysate is provided in figure 1B, we consider this sufficient as both – figure 1A and B – show the reactivity with cell lysate and figure 1A is primarily concerned with the band size, whilst the quantity can then be assessed in figure 1B.

Regarding the ConA precipitation control, we agree with this reviewer that precipitation of a control protein would be the most elegant way to control for successful precipitation. We would however like to draw the attention of this reviewer to the original blots submitted with the manuscript, as the same contaminating bands are present in the ConA precipitate of MDBK and CRIB-1 cells upon longer exposure. Given this evidence in addition to the latter presented genetic evidence and the fact the precipitation was performed in parallel, we consider the results of our ConA precipitation valid even in the absence of a positive control of precipitation. 

Reviewer 2 Report

The manuscript entitled “ADAM17 is an essential factor for the infection of bovine cells with pestiviruses” demonstrates the importance of ADAM17 for pestivirus infection based on the resistance of CRIB cells to BVDV infection. The authors provide interesting and relevant results, contributing to the pestivirus biology understanding. The manuscript is well written and complete. 

Minor comments:

1-The use of BVDV3 does not seem to be the most acceptable/used nomenclature. Please consider using HoBi-like pestiviruses (HoBiPev) to avoid further confusion in the classification.

2-Line 103, please add the HoBiPev strain name and check if the references 42 and 43 were correctly added. In the same portion of the text, I would suggest the authors to include the subtype for each virus described (when/if available). 

Author Response

Please find the comments of reviewer 2 below in black font, our answers in blue and changes to passages in the text in blue and italics. 

The manuscript entitled “ADAM17 is an essential factor for the infection of bovine cells with pestiviruses” demonstrates the importance of ADAM17 for pestivirus infection based on the resistance of CRIB cells to BVDV infection. The authors provide interesting and relevant results, contributing to the pestivirus biology understanding. The manuscript is well written and complete. 

We sincerely thank this reviewer for the positive feedback on our manuscript and the below provided suggestions!

Minor comments:

1-The use of BVDV3 does not seem to be the most acceptable/used nomenclature. Please consider using HoBi-like pestiviruses (HoBiPev) to avoid further confusion in the classification.

We thank the reviewer for pointing this out and have adjusted the nomenclature to HoBi-like pestivirus throughout the manuscript.

2-Line 103, please add the HoBiPev strain name and check if the references 42 and 43 were correctly added. In the same portion of the text, I would suggest the authors to include the subtype for each virus described (when/if available). 

We checked the references, and they are correct as mentioned. The HoBiPev strain employed by us was isolated from a serum contamination as mentioned in ref 43 and the strain is GI2011_1. We added the subtype of the other viruses if known as follows:

Line 102 f: Viruses employed in this study included CSFV Alfort Tübingen [41] (subtype 2.3 [42]), Linda virus (LindaV) [43], a HoBi-like pestivirus (HoBiPeV) strain (Gi2011_1) isolated from a serum contamination [44] and the BVDV strains NADL [4] and C87 (both subtype 1A [4]).